# An Overview on the Novel Core-Shell Electrodes for Solid Oxide Fuel Cell (SOFC) Using Polymeric Methodology

**DOI:** 10.3390/polym13162774

**Published:** 2021-08-18

**Authors:** Rong-Tsu Wang, Horng-Yi Chang, Jung-Chang Wang

**Affiliations:** 1Department of Marketing and Logistics Management, Yu Da University of Science and Technology, Miaoli County 36143, Taiwan; rtwang@ydu.edu.tw; 2Department of Marine Engineering (DME), National Taiwan Ocean University (NTOU), Keelung 202301, Taiwan

**Keywords:** intermediate temperature solid oxide fuel cell, interface charge transfer impedance, diffusion impedance, core-shell structure, triple-phase boundaries, electrode electrocatalytic activity

## Abstract

Lowering the interface charge transfer, ohmic and diffusion impedances are the main considerations to achieve an intermediate temperature solid oxide fuel cell (ITSOFC). Those are determined by the electrode materials selection and manipulating the microstructures of electrodes. The composite electrodes are utilized by a variety of mixed and impregnation or infiltration methods to develop an efficient electrocatalytic anode and cathode. The progress of our proposed core-shell structure pre-formed during the preparation of electrode particles compared with functional layer and repeated impregnation by capillary action. The core-shell process possibly prevented the electrocatalysis decrease, hindering and even blocking the fuel gas path through the porous electrode structure due to the serious agglomeration of impregnated particles. A small amount of shell nanoparticles can form a continuous charge transport pathway and increase the electronic and ionic conductivity of the electrode. The triple-phase boundaries (TPBs) area and electrode electrocatalytic activity are then improved. The core-shell anode SLTN-LSBC and cathode BSF-LC configuration of the present report effectively improve the thermal stability by avoiding further sintering and thermomechanical stress due to the thermal expansion coefficient matching with the electrolyte. Only the half-cell consisting of 2.75 μm thickness thin electrolyte iLSBC with pseudo-core-shell anode LST could provide a peak power of 325 mW/cm^2^ at 700 °C, which is comparable to other reference full cells’ performance at 650 °C. Then, the core-shell electrodes preparation by simple chelating solution and cost-effective one process has a potential enhancement of full cell electrochemical performance. Additionally, it is expected to apply for double ions (H^+^ and O^2−^) conducting cells at low temperature.

## 1. Introduction

The fuel cell (FC) belongs to an electrochemical reactor that straightly converts the chemical potential of a fuel into electrical energy. Therefore, the FC capability is not restricted by the Carnot cycle count, which is the regulation for some mechanical facilities (e.g., steam turbine or internal cumbustion engine). The power conversion efficiency of a fuel cell is about 45% principally such as proton exchange membrane fuel cell (PEMFC). Taner’s group [1,2] has studied hydrogen gas that entered the PEMFC anode through the catalyst anode to form the proton. The protons pass through electrolyte membrane to generate electricity due to the released electrons passing through external circuit to the cathode side underwent reduction interactions of the oxygen gas. This experimental study proves that the H_2_ pressure drop and water management at cathode side are the important factors to affect the fuel cell performance. If combining the heat recovery system, a solid oxide fuel cell (SOFC) can reach the high power efficiency of 80% [3] with no water treatement issues because of higher 100 °C operation temperatures. Thus, fuel cells possess characteristics of high performance of energy conversion, environment-friendly, multiple choices of fuels and high waste heat recovery rate. Figure 1 displays that the SOFC has the supreme efficiency of power conversion among various fuel cells and demonstrates the SOFC-combined cycle system achieving the highest power generation efficiency, which can be employed in a wide range of power outputs (large-scale power generation of kW~MW). It is indicated in Figure 1 marked with red circles. High power output lets SOFC suite to large power generation capacity systems, for example, various power plants and container ships. The SOFC is also a good candidate for large distributed energy systems [4,5].

There are three consisting parts in a SOFC that are anode, electrolyte and cathode. A schematic diagram of a general planar design of SOFC is shown in Figure 2. The planar cells can be electrolyte-supported, electrode-supported and metal-supported types. Planar patterns provide a number of potential benefits, containing easier and cheaper fabricating procedures and higher power densities than tubular devices [6,7]. Thus, the planar SOFC is an excellent candidate for high power requirement such as power plant, cargo and container ships [8]. The operation of a planar fuel cell stack requires bipolar plates to connect the membrane electrode assembly (MEA) and external circuit or to combine another MEA. The bipolar plates allow fuel and oxidant flow through their channels [9]. The electrochemical capability and performance of planar SOFCs is extremely based on the materials of components, the microstructure of the electrodes and the geometric parameters of the cell [10].

The typical oxygen ion conducting SOFC is Ni/YSZ cermet for anode, 8 mol% Y_2_O_3_ stabilized ZrO_2_ (YSZ) for electrolyte and perovskite La_1−x_Sr_x_MnO_3_ (LSM) for cathode. Such a schematic diagram of SOFC is shown in Figure 3 [11]. The operation mechanism is that the cathode material receives the electrons from the external circuit to reduce oxygen molecules into oxygen ions (O_2_ + 4e^−^ → 2O^2−^) at triple-phase boundaries (TPBs) of cathode/electrolyte; then the O^2−^ is conveyed into electrolyte, the electrolyte material transports the oxygen ions to the TPBs of anode/electrolyte to oxidize the fuel H_2_ (O^2−^ + H_2_ → H_2_O + 2e^−^), which comes from the anode side, then the electrons are transported to the current collector and further to external circuit, which is indicated in left picture of Figure 3. The reaction products are pure water and heat. The electrolyte material also plays a role to prohibit the electrons from the anode through electrolyte into the cathode.

Generally, the operation temperatures are in the range of 800–1000 °C for ZrO_2_-based electrolyte [12], which is known as high temperature solid oxide fuel cell (HTSOFC). In the Ni-YSZ anode system, the reduced Ni metal network acts as electron conducting path and the YSZ can decrease the thermal expansion mismatch between Ni metal and YSZ ceramic electrolyte. The agglomeration and coarsening of Ni particles during SOFC operation at high temperature of 800–1000 °C is a major concern, since Ni particles sintering results in loss of active surface area and decreased conductivity of the anode. When using hydrocarbon fuels in a Ni-based anode of SOFCs, the major concerns are the anode catalyst coking and sulfur poisoning since H_2_S exists in most of fossil fuels. The Ni-based anode catalyst can be easily deactivated by carbon deposits and suffers irreversible sulfur poisoning [13].

Decreasing the operating temperature to below 800 °C (under 600 °C is even better) permits the application of fuels involving methane and butane without pre-forming a hydrogen fuel. The advantages of a lower operating temperature incorporate the wide selections of materials, interface matching and long-term stability of the cell system and cost-effective operation and fabrication [14].

### 1.1. Polarizations of Fuel Cells

The energy conversion efficiency of a SOFC is mainly controlled by interface charge transfer, ohmic and diffusion impedances (or polarizations), a well-known electrochemical performance presented in Figure 4. General SOFC consists of two porous electrodes and an electrolyte assembled as the micrograph shown in the inset of Figure 4. Well-controlled gas diffusion, electron and ion conduction/diffusion are all considered key points in a SOFC operation to obtain a high peak power.

The electrochemical performance of SOFC is greatly determined by the species of H_2_ and O_2_ reaction processes in the porous electrode, such as oxygen adsorption, dissociation, surface diffusion to the TPB, ionization and incorporation into the electrolyte and bulk paths. Fewer TPBs exist at the electrode/electrolyte interfaces. The electrochemically active interface region is expected to extend a few micrometers from the electrolyte to the electrode and plays essential role in terms of the performance and durability of SOFCs [15,16].

Due to the significance of the electrode/electrolyte interface and electrocatalytic activity enhancement required, many studies pay efforts on the extension and enlargement of the TPB area [17,18]. The electrochemical reactions mainly proceed at the TPB sites of gas (H_2_ or O_2_), electrode and electrolyte. Theoretical calculations and experimental results have showed that a composite electrode should exhibit low charge transfer polarization by spreading the electrochemically active area within the volume of the electrode [17,19,20]. For example, the TPB length was measured by multiplying the average length of the cathode particle at the interface with the number of particles per unit area. The results suggested a three-dimensional distribution of TPB in LSM-YSZ composite cathode leads to a significant drop of cell overpotential [21,22]. Therefore, the manipulation of the electrode/electrolyte interface microstructures actually plays a crucial role in determining the overall cell performance and durability.

The interface charge transfer polarization results from TPBs between anode and electrolyte, and between cathode and electrolyte materials. The diffusion polarization originates from the internal structure including grains connection and porosity of an electrode. The ohmic polarization occurs from electrolyte material and external conducting connector structures. One significant bottleneck in the development of an intermediate temperature solid oxide fuel cell (ITSOFC), which is operated at 500~800 °C, is that the solid electrolyte exhibits low oxygen ionic conductivity at such a temperature, e.g., 600 °C. Apart from lowering the thickness of electrolytes, realizing the capability to acquire a high ionic conductivity composition of electrolytes is a major objective for the ITSOFC. It is obvious that the ionic conductivity can be enhanced via ceria doped with appropriate aliovalent cationic dopants. Co-doping has been certified to successfully increase the electrical characteristics of ceria-based electrolytes [23]. The (La_0.75_Sr_0.2_Ba_0.05_)_0.175_Ce_0.825_O_1.891_ (LSBC) is a typical electrolyte for ITSOFCs utilized and developed in our research group [24]. When the ceria-based electrolyte material is ready selected for an ITSOFC, the lowering interface charge transfer and diffusion impedances are determined mainly by the electrode materials selection and manipulating the microstructures of electrodes. Thus, how to extend and enlarge TPBs area are significant topics.

Composite electrodes by mixing ionic and electronic conducting materials are used to improve electrodes’ performance. Such composite electrodes including anode and cathode help to enhance the properties of mixed electronic–ionic conductors and the inter-component compatibility [7,25,26,27,28,29,30,31]. The composite electrodes are good ideas utilized by a variety of mixing and impregnation or infiltration methods to develop efficient anode and cathodes effectively. The mixed electrodes, e.g., Ni/YSZ anode and YSZ/LSM cathode in conventional SOFC extending TPBs, already reduce the interface charge transfer impedance and improve the electrochemical performance, as the schematic electrode microstructures shown in Figure 3.

### 1.2. Conventional TPBs Extension in Electrodes

In order to improve the stability and activity of the electrode/electrolyte interface, electrode modifications such as impregnation or infiltration can be regarded as an instrumental pathway. By infiltrating a layer of PrBaMn_2_O_5__+__δ_ (PBM) into BaZr_0.1_Ce_0.7_Y_0.1_Yb_0.1_O_3__−__δ_ (BZCYYb) electrode, a fuel cell substantially enhanced stability and performance at reduced temperatures [32].

The typical electrode TPBs extension microstructure of La_0.75_Sr_0.25_Cr_0.5_Mn_0.5_O_3−δ_ (LSCrM)-impregnated anode has been manufactured via infiltrating 70% porous yttria-stabilized zirconia (YSZ) matrixes with an LSCrM solution. The LSCrM is a principal electron conductive phase while the well-sintered YSZ porous anode supplies an ionic conduction route in every part of the electrode. Further, Silver (Ag) and Nickel (Ni) are complemented by nitrate impregnating methods for improving electronic conductivity and electrocatalytic activity. The various impregnated microstructures are shown in Figure 5 [33].

The interface polarization may be caused by a non-conducting phase at the interface resulting from the solid-state reaction between electrolyte and electrode. Control the interface reaction and extend the TPBs are both considering topics. A typical composite cathode is the adherence of meticulous SDC particles to the surface of crude BSCF cathode particles, which resulted from the mechanical admixture of Ba_0.5_Sr_0.5_Co_0.8_Fe_0.2_O_3−δ_ (BSCF) + Sm_0.2_Ce_0.8_O_1.9_ (SDC) (70:30 in weight ratio). The phase reaction can contract merely at the interface between BSCF and SDC and insignificantly dominant in particles internal parts, thus, everywhere the whole BSCF cathode particles, individually, with the formation of a thin layer of novel (Ba, Sr, Sm, Ce) (Co, Fe)O_3−δ_ perovskite phase at a firing temperature as has been shown in Figure 6. The fine SDC particles enclose the BSCF coarse particle and maintain a single phase of SDC and BSCF, individually. The inter-phase intimately adheres BSCF and SDC to reduce the interface polarization. Thus, the sintered BSCF + SDC electrode shows an area specific surface resistance reduction above the BSCF because of the amplified cathode surface area promoted by the meticulous SDC particles. An improved peak power density at 600 °C was achieved for a thin film of the electrolytic cell with the BSCF + SDC cathode fired from 1000 °C [34].

Other composite electrode examples are also presented as follows. The Ni-BZCY/SDC/BSCF cell with an interfacial reaction that can be manipulated to form a secondary phase at anode Ni-BZCY and SDC with an electronic conductor to benefit cell performance and power output [35]. The secondary phases nickel aluminate spinel (NiAl_2_O_4_) and zirconium titanite (Zr_5_Ti_7_O_24_) formation by infiltrating a small amount of aluminum titanite (ALT, ~4 wt%) into the Ni-YSZ anode scaffold were found to suppress Ni coarsening and expand the electrode’s TPBs [36,37].

Another TPBs extension method is an introduction of the anode functional layer (AFL) with fine microstructure at the anode/electrolyte interface to increase the TPB length and to restrain the activation polarization for hydrogen oxidation [38,39,40]. This kind of technique meets a trade-off between the enhancement of electrochemical performance due to the increasing TPB and the decrease in performance due to the increased gas diffusion resistance.

### 1.3. Proposed Core-Shell Electrodes

The SOFC synthesis needs high co-firing temperatures (often above 1000 °C) as to result in chemical reactions between the perovskite-based electrode, such as LSM, and the zirconia-based electrolyte; the formation of a secondary phase at the interface is usually insulating, and thus may impact the stability and performance of a SOFC [16,41,42,43,44].

In the conventional impregnated or infiltrated solution into the porous electrode scaffold, the nanoparticles are formed over the electrode scaffold surface at a relatively low temperature, necessarily (<800 °C), as the other way, it is not reactive when sintering under such temperatures. Therefore, the isolated nanoparticles constituted on the scaffold have comparatively disappointing electrical conductivity as a result of the lack of reactive sintering of a normal infiltration–sintering process. In order to fill nanoparticles to cover the scaffold surface enough, the impregnation or infiltration process is required to repeat several times [45]. Such a repeated process often causes the block of porosity in the electrode and inhibits the thorough impregnation into the electrode/electrolyte interface. If the reactive temperature is so high as to enhance the reaction of impregnated nanoparticles and electrode grains, further coarsening or necking the impregnated nanoparticles or electrode grains significantly [33,46,47,48]. The particles agglomeration occurs to reduce the electrode porosity. Secondary phases are also generated due to the diffusion of impregnated nanoparticle composition into the electrode lattice seriously. The porosity reduction and too significant secondary phase existence will affect the TPBs function and contribute the polarization increase, also resulting in a coefficient of thermal expansion (CTE) of electrode mismatching with the electrolyte.

A core-shell structure is pre-formed during the preparation of electrode particles. A high enough amount of shell nanoparticles forming a continuous charge transport pathway increases electronic and ionic conductivity of the electrode. The electrode activity is then improved as contrasted with mechanical mixing composite electrodes and traditionally impregnated electrodes. Furthermore, the core-shell electrode configuration can successfully enhance the thermal stability by preventing more sintering and thermomechanical stress because of the CTE matching with the electrolyte. In recent decades, the core-shell electrodes of anode and cathode were developed by our research group to contribute the extending TPBs and solve the mismatching issue of the electrode and electrolyte. Significant improvement in electrocatalytic performance and in impedance have been achieved.

The core-shell is not a new terminology. It has been used for ceramics in several studies [49,50,51]. However, in our work, the core-shell electrodes pre-formed by chelating solution to extend TPBs were appreciated as a beneficial developing technology in future for SOFC [13]. The main objective of this work was therefore to demonstrate the progress of core-shell electrodes by our efforts to provide a feasible, convenient, cost-effective and time saving TPBs extension technique. The proposed core-shell electrodes, from fabrication by chemical chelating and solution coating processes to electrical and electrocatalytic characterization, are informed in the later parts. Furthermore, the recent developing pseudo-core-shell anode by thin electrolyte impregnation is introduced and compared with other ITSOFC research. The pseudo-core-shell or inverse impregnation is further expected to apply to double ions (H^+^ and O^2−^) conducting low temperature fuel cells.

## 2. Core-Shell Electrodes Preparation

### 2.1. Core-Shell Anode Preparation

Anode materials must suffice a number of demands involving enough ionic and electronic conductivities, thermal compatibility, outstanding electrocatalytic (electrochemical oxidation of hydrogen) activity and stable chemistry. An anode utilizes a ceramic composed of a mixing ionic and electronic conductors (MIEC) can enhance the ionic conductivity to the valid range of the TPBs and provides relative electronic conductivity. The perovskite ABO_3_ structure is a good candidate of MIEC because it adapts to space and stoichiometric bias owing to doping different ions with distinct valences to either increase the conductive and catalytic activity of tolerant sulfur, or to expand steadiness with productive electrochemical function. The MIEC materials can solve the issues with the commonly used anode material, Ni/YSZ cermet, which exhibits wonderful electrocatalytic characteristics for the gathering of the beating current and fuel oxidation, but it reveals some weaknesses containing the liable sulfur toxin, carbon depositing, growing of Ni-particle and unstable volume in reduction–oxidation (redox) cycling under H_2_/H_2_O atmospheres [12,52]. The specifically beneficial characteristics of ABO_3_ are delivering electrons that jump between mixing valent cations and capturing the conduction band as to raise the conductivity through the donors doped on A-site or the 4d or 5d transition ions doped on the B-site [53,54].

Our work proposed a structure of a core-shell anode that constituted of a core of conducting perovskite and a electrocatalytic shell of CeO_2_-based electrolyte. The benefits of core-shell structure include a simple anode material and structure preparation without the tedious mixing or impregnation of second functional particles to improve conductivity, electrocatalytic activity and thermal matching with the electrolyte. The overall heat treatment cycles may be decreased. The ABO_3_ structure of (Sr_0.7_La_0.3_)(Ti_0.9_Nb_0.1_)O_3_ (SLTN) anode reveals relative conductivity in reducing atmospheres. The multiple-elements doped ceria-based electrolyte (La_0.75_Sr_0.2_Ba_0.05_)_0.175_Ce_0.825_O_1.89_ (LSBC) displays superior ionic and electrocatalytic attributes at intermediate temperatures [24,47,55]. The core-shell particles SLTN-LSBC as anode materials were constructed, and electrical characteristics of anodes were characterized [56].

The anode core was prepared by a ball-milled method using La_2_O_3_, SrCO_3_, TiO_2_ and Nb_2_O_5_ as the starting materials. The powders of ball-milled SLTN were calcined at 1100 °C-4 h in air. The raw materials of La(NO_3_)_3_·6H_2_O, Sr(NO_3_)_2_, Ba(CH_3_COO)_2_ and Ce(NO_3_)_3_·6H_2_O were employed in the citric acid-based solution (SV) combustion skill to provide the coating shell. These La, Sr, Ba and Ce salts were dissolved in de-ionized water to constitute an aqueous solution. Before combustion, The SV was composed of a molar ratio of 1:2 for LSBC:citric acid for the mixed aqueous solution. The citric acid (C_6_H_8_O_7_) contains three COO^−^ chelating ligands, which can chelate metallic ions to facilitate the homogeneous formation of shell composition during the combustion process. Hydroxyl groups of polysaccharides can be modified by chemical modification such as the thermal gelation method [57,58,59]. The expected homogeneous shell coating on our proposed electrode core particles is the same objective as the homogeneous distribution and the absence of agglomeration reported by the chitosan/pectin polymeric matrix to prevent the formation of nanoparticle clusters [60,61].

The calcined core powders of SLTN were put into the citric acid-based LSBC solution, which the molar ratio was SLTN:LSBC = (100 − x):x, where x was 0.75, 1.5, 3, 6 or 12. Subsequently, the chemical mixing solution was heated on a hot plate at 90 °C to evaporate the water matter, which transformed into a yellow gel, afterward, which was oven-dried at roughly 100 °C. Accordingly, the dried gel was crushed and calcined at 900 °C for 2 h in the air to fabricate the powders of the core-shell anode. The as-prepared core-shell anode powders were termed as SLTN-x mol% LSBC. A single anode disk was subsequently manufactured by a uniaxial press in a die from all prepared powders of SLTN-x mol% LSBC that was granulated with a binder. The alone anodes were consequently sintered at 1300 °C for 3 h in an activated carbon-reduction atmosphere.

### 2.2. Core-Shell Cathode Preparation

The mostly studied perovskite cathode materials contained (La, Sr)MnO_3_ (LSM); (La, Sr)(Co, Fe)O_3_ (LSCF); and (Ba, Sr)(Co, Fe)O_3_ (BSCF) series. The LSM material had excellent co-fired matching with a YSZ or ceria-based electrolyte in HTSOFC. Nevertheless, the electron-conductivity of LSM decreased as it was manipulated at an ITSOFC due to lowering temperature. Perovskite LSCF and BSCF with MIEC characteristics have longer-term of TPBs than electron-conductive LSM [62,63]. The electrocatalytic activity of oxygen in LSM is poorer than that in LSCF and BSCF. However, the CTE for both LSCF and BSCF are about 20 × 10^−6^ K^−1^, which are fairly greater than that of the ceria-based electrolyte (~12.5 × 10^−6^ K^−1^). The present thermal mismatching trouble brings about the questions in the operating temperature of the fuel cell and in co-firing with the electrolyte. Furthermore, there are still having many disadvantages for Co-based cathodes containing the large evaporation owing to the reduction of Co, high cost and the transition of Co^3+^ (with octahedral coordination) from a low- to a high-spin states [64,65,66]. Ferrate-based materials without cobalt are well candidates for the cathode material since the iron is cheap and reveals nearly zero toxicity. The Ba_0.5_Sr_0.5_FeO_3−δ_ (BSF) has the supreme electrical conductivity among ferrates [67,68] but this perovskite oxide still displays a large CTE [69]. Accordingly, it is difficult to apply for co-firing with ceria-based and zirconia electrolytes directly.

The iron perovskites display prominent conductivities of oxygen ions resulting from the lower B–O bonding energies and the facile transitions of charge carriers between the various coordination polyhedral. They are easily collapsed by the moisture from the air, also unstable at high temperatures and low partial pressures of oxygen [70]. The Ce doping in BSF provides lattice stability, better cathode–electrolyte adhesion and enhanced cell performance by increasing the TPBs in SOFC [71,72].

The as-prepared BSF particles were coated by the Ce component by utilizing an ethanol-water mixed semi-organic solution (SOS). A similar solvent removal and decomposition method was utilized in the production of hydrocolloid film [73]. The Nb was used to modify the B-site of BSF, referred to as BSNF, to enhance it with structural and environmental stability. Subsequently, the La and Ce elements were coated onto the as-prepared BSNF particles also employing an ethanol–water mixed SOS. The stability of the BSF, BSNF structure and the obtained electrical properties were surveyed with respect to the influences of the Ce, La diffusion into the BSF and BSNF [72].

The calcined and pulverized BSF and BSNF powders were dispersed in absolute ethanol, individually. The aqueous solution of 20 mol% La(NO_3_)_3_·6H_2_O and 80 mol% Ce(NO_3_)_3_·6H_2_O was prepared in DI water, referred to as LC. The La(NO_3_)_3_·6H_2_O aqueous solution or Ce(NO_3_)_3_·6H_2_O aqueous solution or LC solution was added to the dispersed BSN and BSNF. Hereafter, the volume ratio of water to ethanol in every one reaction mixture was regulated to 1:9. Such SOS containing Ce, La and LC to coat the powders of BSF and BSNF were hence referred to as BS(N)F-y Ce (y = 2.5, 5 and 10 mol%), BSNF-y La and BSNF-y LC (y = 1.25, 3.75, 5, 10, 15, or 25 mol%), respectively, then each was stirred for 6 h and then dried at 80 °C with successive stirring. The powders were later ground and then calcined under 800 °C for 4 h.

The terpineol and ethyl cellulose were further mixed with these calcined powders including the BSF, BS(N)F-y Ce, BSNF-y La and BSNF-y LC. Another powder of the calcined LSBC was mixed with the binder named PB72. The ball-milling machine was employed to homogenize each binder-containing powder. After drying, these powders were granulated utilizing the mortar with a pestle and transited through a screen of 60 mesh. Each of the granulated powders was then pressed in a die employing a uniaxial press to form a disk. The pressed disks were next subjected to traditional sintering in an electric furnace (6 h at 1150 °C for BSF, BS(N)F-y Ce, BSNF-y La and BSNF-y LC electrodes; and 6 h at 1500 °C for LSBC electrolyte).

## 3. Results and Discussion of Core-Shell Electrodes

### 3.1. Core-Shell Anode of SLTN-LSBC

If a low ratio of LSBC (e.g., 1.5 mol%) displayed a shell, then the amounts of nanoparticles of LSBC were not clear enough to recognize. Figure 7a shows that the larger particles of the calcined SLTN core were adhered by enough distinguished LSBC nanoparticles (12 mol% LSBC) [56]. The elemental analysis on the core-shell SLTN-12 mol% LSBC powders is easily identified as the inserted EDS spectrum shown in Figure 7a. The primary components of Ti, Ce and Sr were mapped on the morphology of Figure 7a obtained from the EDS elemental analysis. Figure 7c demonstrated the Ce element existed on the nanoparticles of the shell, which further certified the core-shell formation. The Titanium (Ti) and Strontium (Sr) assigned over core-shell major body as shown in Figure 7b,d. The Ti and Sr mapping images demonstrated the core SLTN existed below the LSBC nanoparticles shell.

The conductivity contribution of La and Nb donors-doped perovskite structure was enhanced if the SLTN sintered in the reduction atmosphere. Figure 8 exhibits that the DC conductivity decreased with the increase in the molar ratio of shell LSBC, which demonstrated that the the resistance of shell LSBC affected the electrons hopping as the ratio of shell LSBC increased. The more shell nanoparticles coated on the core, the more favorable its role for ionic conductivity was. A ceria-based shell may increase the electrocatalytic activity but it would lower the electronic conductivity. The lesser shell coating such as 1.5 or 3.0 mol% LSBC shown in Figure 8 maintains appropriate high electronic conductivity and electrocatalytic activity simultaneously. This situation is the electron transfer at the TPB of the electronic core and ionic shell [24,47,55]. Ce^3+^ was possibly formed at the anode in a reduction atmosphere at the higher temperature, 700 °C, in Figure 8 and enhanced the electronic conductivity [74].

Such few shell coatings less than 3 mol% also altered the peak semiconducting activity to a lower critical temperature (T_c_) of 500 °C. Figure 8 exhibits that the conductivity of metallic behavior of the core-shell anode as the measuring temperature is larger than the T_c_. The present achievement further proved that the shell of ion may shift the redox reaction to a lower operating temperature than the one without coating ionic shell. Figure 8 reveals that a high covering ratio of shell on core (e.g., 6.0 or 12.0 mol% LSBC) displays a degraded electrical conductivity on the core-shell anode. The lattice oxygen led to the coexistent generation of oxygen vacancies and Ti^3+^ ions at a high temperature and low oxygen partial pressure. More oxygen vacancies produced from a high covering ratio of the ionic shell such as 12 mol% did not distinctly contribute to the overall DC conductivities of the SLTN core [54] as to lower the total conductivities of the core shell anode and owing to the low mobility of oxygen vacancies contrasted with electrons at a temperature of higher than 700 °C.

In addition, measuring the AC impedance of the core-shell anode may provide testimony of the ionic shell of LSBC to enhance the electrocatalytic activity and electron transfer rate. Figure 9 exhibits the AC complex impedance analyses. Those indicate that the LSBC shell (x = 0 to 3.0) profited the impedance decrease of the core-shell anode. However, the larger covering ratio (x = 6.0) of the LSBC shell increased the impedance of the core-shell anode. This result responds to the decrease in DC conductivities for the larger amount of shell coatings. AC impedance data acquired in Figure 9 may be roughly adapted as two depressed impedance semicircles for each anode sample according to RQ equivalent circuits containing parallel resistance/CPE circuits in series [24,56,75,76,77,78,79,80,81]. The SOFC is full solid state device, in order to corresponding to the illustrations in Figure 4, the detail electric double layers are not considered and discussed in the work. The depressed arc at a high-frequency range is concerned with the interface charge transfer process, while the arc at a low-frequency range is related to surface diffusion processes and hydrogen dissociation. The three intersections on the ReZ axis from left to right for each x mol% LSBC shell coating express the ohmic resistance (R_e_), interface resistance of charge transfer (R_i_), and electrocatalytic resistance having no electron transfer (R_c_), respectively. The valid resistance dedication is indicated as ReZ(d) = R_c_ − Ri for diffusion polarization and the interface charge transfer polarization of ReZ(i) = R_i_ − R_e_. In addition to the higher than 6.0 mol% LSBC shell coating, the LSBC shell coating decreased ReZ(d) effectively and increased the ReZ(i) slightly. The present work illustrates the core-shell structure with low LSBC coating increased the electrocatalytic activity of anode due to the extension of TPBs.

### 3.2. Core-Shell Cathode of BSNF-LC

Although Abd Aziz et al. [7] have reviewed composite cathode materials on the addition of SDC or GDC electrolyte material to traditional cathode materials recently with relative progress to operate SOFC at intermediate to low temperatures, our core-shell technique is still relevant and exciting to study.

In our study [72], the BSNF coated by LC moderately decreased the core size and enhanced the porosity as shown in Figure 10a–d. Figure 10b exhibits the evident shell particles on the BSNF core resulting from La coating after sintering. Compare Figure 10b with Figure 10c,d, it also shows that the existence of Ce inside the LC coating layer promoted the diffusion of La into BSNF, according to 10LC, overcoming the limited solubility of La. The comparison in Figure 10c,d reveals a tiny morphological change for adding the LC coating between 5 and 10 mol%. The diffusion of La into BSNF constructed an LC layer as to increase the TPBs. Figure 10e, the EBSD micrograph of BSNF-5 LC, obviously illustrates that the LC diffused into the BSNF structure and an LC layer (red dots) developed near the core boundaries of BSNF. Moreover, precipitation was observed as high amounts of LC coating were applied, as shown in Figure 10d. No second phase was discovered in BSNF-3.75 LC or BSF-5 Ce under XRD resolution. It indicated that the Ce could enhance the La solubility into BSNF but a small amount of 3.75 mol% LC cannot change the structure of BSNF or a second phase to be detected. The Ce coating layer decreased the core size and enhanced the porosity of BSF, whereas based on the relative densities and microstructures, the LC coating layer imposed tiny influences on porosity or core size for BSNF.

Izuki et al. [82] utilized a diffusion couple prepared by depositing LSCF thick film onto the GDC substrate using Pulse Laser Deposition (PLD) technique to study the LSCF/GDC interface reaction. The diffusion couple was then treated at 1000–1200 °C up to 672 h, and a significant diffusion of La into GDC as well as the diffusion of Ce into LSCF were detected. The 3.75 mol% LC shell coated on BSNF under heat-treating at 1150 °C/6 h with little diffusion was then considered reasonably.

Figure 11 demonstrates the conductivities of the BSF (black solid line) with similar values to those of the BSNF (black dashed line). Their peak conductivities were the same, at about 510 °C. Fe cations in ferrate perovskite are in mixed Fe^3+^/Fe^4+^ valence states [83,84], and the decrease in the electrical conductivity at high temperatures exhibited by ferrate perovskites (such as BSF) is attributable to the release of lattice O and the Fe ions’ spin transitions [85]. The DC conductivity of BSF and BSNF reveals that Nb doping did not improve the conductivity of BSF, although it stabilized the perovskite structure. Figure 11 shows that the Ce coating 5 and 10 mol% increased the BSF’s conductivity but also increased its transition temperature to above 510 °C. The presence of Ce in the BSF lattice may reduce the Fe ions’ spin transitions by compensating for Fe’s high valence state. Ce may also suppress the loss of lattice O and modify the electronic conduction [86]. The transition temperature (T_c_ ~ 510 °C) from semiconductor-like to metal-like conductivity for BSNF-3.75 LC (open circle red line) was lower than those of samples exceeding 5 mol% LC-coating. The lower LC-coating concentration on BSNF could enhance its conductivity and reduce the operating temperature. In contrast, using a high LC concentration (e.g., 10 mol%) provided excessive ionic conduction but did not effectively reduce the transition temperature or improve the conductivity.

The mixed doping at both the A- and B-sites, such as Ba, Sr for A-site and Nb, Fe for B-site, is important for maintaining the oxides’ disordered oxygen vacancy structures [67,87]. The excessive Ce coating on BSF-10 Ce generated a second phase, in which the high porosity and small grain size reduced the Ce-doping effect by increasing the oxygen-vacancy clustering. These effects may delay the metal-like transition and elevate the transition temperature of peak conductivity, as shown in Figure 11. The diffusion of Ce into the BSF lattice induced stress. Extending TPB is believed to enhance the small polaron hopping mechanism and improve the conductivity.

Similar to the core-shell anode impedance analyses in Section 3.2, the AC impedance data of BSNF/LSBC/Pt, (BSNF-5La)/LSBC/Pt and (BSNF-3.75LC)/LSBC/Pt half-cells (Figure 12) also can be approximately fitted to two depressed impedance semicircles according to the RQ equivalent circuits [24,72,75,76,77,78,79,80,81]. The depressed arc in the intermediate frequency range is still related to the interface charge-transfer process, diferently at low frequencies instead of H_2_, the arc is associated with O_2_ dissociation and surface diffusion processes on the core-shell cathode grains. Therefore, the effective resistance is ReZ(i) = R_i_ − R_e_ for the interface charge transfer and ReZ(d) = R_c_ − R_i_ for chemical catalysis reaction. The chemical catalysis enhancement is largely attributable to the decrease of ReZ(d) in the half-cells containing BSNF-x LC core-shell cathode. These half-cells also exhibit significantly decreased interface resistance ReZ(i) and ohmic resistance R_e_.

The composite cathodes used for IT-SOFCs have been proven as an excellent technique to achieve low polarization and high power. The SDC or GDC electrolyte mixed into LSCF as composite cathodes are well studied [45,88,89,90]. The benefits are attributed to the enlargement of the TPBs and the excellent adhesion between the electrode and electrolyte by adding electrolyte species to the cathode material. However, the cobalt content is expensive and evaporates during processing and operation. Those are all significant issues when using cobalt-based cathodes. Additionally, the mixing or infiltration amount of electrolyte cannot be controlled precisely in every repeated infiltration process from porous cathode outside surface into cathode inner space sufficiently or in random mixing process that affects the quality of the composite cathode. Generally, a large amount of 50 wt% electrolyte addition [45] was also necessary to obtain the best cathode properties.

The cell reliability depends on the structural stability and optimal CTE matching among the various interfaces. The BSF cathode-LSBC electrolyte interface was observed to exhibit peeling after two cyclic tests. The CTE values of the LSBC electrolyte substrate and various electrode materials, including uncoated BSF and BSNF and shell coated BSF-5 Ce and BSNF-3.75 LC, were measured by a thermal mechanical analyzer (TMA). The linear expansion deviated from referred LSBC (black line in Figure 13) above 450 °C. The linear expansion of BSNF is almost the same as LSBC below temperature 450 °C. Higher than 450 °C, the linear expansion of BSF, BSF-5Ce and BSNF are far larger than LSBC. The BSNF-3.75 LC indicates the smallest deviation from the CTE of electrolyte LSBC.

The CTEs of BSNF-3.75 LC and LSBC below 650 °C with similar relatively could facilitate good interface matching between the cathode and electrolyte while co-firing and operation. The BSNF-3.75 LC half-cell exhibited high power density compared with the uncoated BSNF cathode at 650 °C. This suggests that the LC coating enhanced the power density of the BSNF/LSBC half-cell more significantly than the Ce coating did for the BSF/LSBC half-cell. The core-shell particle-coating by SOS presented in our report could provide a novel approach for achieving high SOFC performance at a low coating amount and intermediate operation temperature.

### 3.3. Psudo-Core-Shell Anode by Ultrasonic Spray Pyrolyzed Impregnation

An La_0.3_Sr_0.7_TiO_3_ (LST) without Nb addition was prepared like SLTN in Section 2.1 and sintered at 1300 °C/6 h to form an approximately 500 μm thick porous substrate. The LSBC electrolyte with polyethylene glycol (PEG) as a chelating agent was prepared in an aqueous solution with a concentration of 0.01–0.1 M. via an ultrasonic nozzle that atomized nano-sized droplets of the LSBC solution by carrier gas into a hot zone. The precursor droplets were decomposed and deposited on a porous LST anode substrate at a temperature of 250–500 °C in the hot zone. An LSBC thin layer electrolyte was successfully deposited on and impregnated into porous LST anode support to form LST/iLSBC half-cells after co-firing at 1350 °C/6 h. This LSBC impregnation and droplets decomposition in porous LST created core-shell like structure, also termed as pseudo-core-shell or reverse impregnation structure. Continuous ultrasonic spray pyrolysis densified a 2.75 μm thin layer LSBC using 0.1 M precursor concentration, 450 °C substrate temperature, 2 mL/min solution flow rate and 25 mL spray volume [91].

Instead of anode function layer (AFL) [92], our ultrasonic spray pyrolyzed pseudo-core-shell anode substrate improved contact and adhesion between the anode and the electrolyte. The dense electrolyte with a thickness of 2.75 μm, as shown in Figure 14a. The profile of the densified 2.75 μm-thick layer of LSBC (iLSBC) on the porous LST substrate is indicated by the EDS line-scanning analysis also shown in the Figure 14a. The top dense layer showed a clear Ce signal (red color) while a trace Ce signal was detected in the LST substrate region over 30 μm depth. The Ce impregnation into the porous LST anode effectively extended the TPBs from the interface between LSBC and LST into the porous LST inner structure. The LSBC impregnation into porous LST is also further identified by electron backscatter diffraction (EBSD) analyses [91].

Refer to LSTN-LSBC and BSNF-LSBC contained half-shell analyses, LST/iLSBC half-cell has approximately fitting to one ohmic resistance and two depressed impedances according to the RQ equivalent circuits, as shown in Figure 14c. This is equivalent to two parallel resistance (R)/constant phase element (CPE) circuits in series with an ohmic resistance of R_e_ [75,76,77,78,79,80,81,91]. The AC impedances of LST/iLSBC half-cell decreases with the measured temperatures increase and diffusion polarization at low frequency decreases significantly as shown in Figure 14b,c. The ohmic resistance (R_e_) and diffusion impedance ReZ(d) significantly decreases and the decrease in ReZ(d) results in an increase in the equivalent capacitance. The impedances ReZ(i) and ReZ(d) of the fuel cell decrease indicated the extension of TPB sites due to electrolyte LSBC impregnation into the porous anode to become a pseudo-core-shell inner coverage. This results in more effective charge transfer and electrocatalytic activity due to longer TPB length and larger area.

The porous microstructure of SOFC electrodes correlates with the TPBs, gas diffusion and concentration polarization. Adding an interlayer such as an AFL between the electrolyte and anode, and impregnating electrocatalytic particles into the porous anode effectively increased the TPBs and decreased the contact resistance between the electrolyte and the electrode [93,94,95,96]. However, the AFL and solution impregnation require additional processing and cost. Repeated impregnation by capillary action may decrease the efficiency of the reforming and electrocatalysis, possibly by hindering and even blocking the fuel gas path through the porous anode structure due to the serious agglomeration of impregnated particles [97]. The ultrasonic spray pyrolyzed thin layer electrolyte could impregnate into the porous anode during deposition to create intimate contact between the anode and electrolyte and act as a dense electrolyte on a highly porous anode support. The sound effects extended TPBs and adhesion of the thin layer electrolyte, enhancing the charge transfer and decreasing the gas diffusion resistances, as the evidence shows in Figure 14.

Our ultrasonic spray pyrolysis pseudo-core-shell deposition technique, using chelating solution precursor, does not require an additional loading press and low pressure or vacuum system. It is operated in an ambient atmosphere with easy adjustment of processing parameters, reduction of synthesis temperature and process cost, as well as convenient operation. The Table 1 is our fabricated half-cell by ultrasonic spray pyrolysis in comparison with other reference anode supported full cells. In Table 1 [98,99,100,101,102,103,104,105], catalytic metals such as Ni or Ru are required in porous anodes for electrocatalytic activity and electronic conductivity enhancement for cells. The LSBC impregnated into LST from the LSBC electrolyte side and sequent formed dense LSBC thin layer electrolyte satisfying the electrocatalytic activity and conductivity requirements simultaneously. Our ultrasonic spray pyrolysis technique has a lower electrolyte preparation and phase formation temperature of 450 °C. The half-cell consisted of 2.75 μm thickness thin electrolyte with pseudo-core-shell anode could provide a peak power of 325 mW/cm^2^ at 700 °C, which is comparable to other reference full cells’ performance at 650 °C. The ultrasonic spray pyrolysis for pseudo-core-shell anode also extended TPBs, enhanced the electrolyte adhesion on porous anode, and deposited dense thin layer electrolyte, all in one process for a half-cell.

In Table 1, Dogdibegovic et al. [98] used higher conductivity SCSZ electrolyte and backbones instead of YSZ, and conventional LSM cathode and SDC-Ni anode were replaced with Pr_6_O_11_ cathode and higher Ni content in anode. The power density of 1.56 Wcm^−2^ was achieved at 700 °C. Such a result is the highest power density up to date using an H_2_-air system and almost approaches to the theoretical OCV value. However, the control of optimizing synthesis parameters is critical and must be careful concerning issues. For the comparison of similar electrolyte thickness (2.08 μm) using repeated spray pyrolysis process on Ni + YSZ anode [105] at 600 °C in Table 1, our future work will likely include spray pyrolysis equipment improvement, anode particle modification and full cell preparation for this anode supported half-cell to further enhance the electrochemical performance. In this overview, we demonstrate only half-cell with (pseudo-) core-shell electrode to achieve relative power density due to extending TPB_S_ with simple, convenient, unexpansive and environmentally friendly technique. The full cells are under-working to promote the power density and study the core-shell interface electrocatalytic mechanisms. Our studies are further expected to apply to double ions (H^+^ and O^2−^) conducting low temperature fuel cells in the future.

## 4. Conclusions and Future Respective

The progress of core-shell electrodes in our efforts is demonstrated with an overview. The core-shell electrode particles and pseudo-core-shell structure are prepared by LSBC electrolyte chemical chelating solution coating and reverse impregnation processes in the anode and cathode. The pre-formed core-shell structure in electrodes, including anode and cathode, provide a feasible, convenient, cost-effective and time saving TPBs extension technique.

The prepared anode SLTN and cathode BS(N)F coarse particles were coated with LSBC electrolyte chelating solution to form core-shell structure. For SLTN-LSBC core-shell anode, the sufficient DC conductivity at intermediate temperature only needed a small amount of 1.5 or 3.0 mol% LSBC shell coating on SLTN. Too molar ratio of LSBC shell was not necessary. Similarly, the half-cell of (SLTN-x LSBC) with x < 3.0 mol% exhibited the decrease in interface charge transfer polarization (ReZ(i)) and chemical electrocatalysis polarization (ReZ(d)) effectively.

The La and Ce formed LC shell coating on BSNF electrode particles increased the peak conductivity and achieved the transition temperature of metal-like conductivity to 510 °C. This is beneficial for the function of a BSNF-LC cathode using in ITSOFC. A low amount of 3.75 mol% LC coating on BSNF reduced AC impedance due to the LC electrocatalytic thin shell influence on the core surface of BSNF, which provided ionic conductivity enhancement among TPBs. The LC coating on BSNF particle also enhanced its CTE matching with LSBC electrolyte. The half-cell of (BSNF-3.75 LC)/LSBC/Pt displayed a higher power density than those utilizing the BSNF or BSNF-5 Ce cathodes at 800 °C. The (Ba, La, Ce) and Nb replaced the A- and B-sites of strontium-ferrate could enhance the more stable BSNF perovskite structure.

The ultrasonic spray pyrolyzed LSBC electrolyte droplets impregnated into LST porous anode to form a pseudo-core-shell and continued the deposition as to resulting in a fired thickness of 2.75 μm dense LSBC electrolyte. This pseudo-core-shell anode substrate improved contact between the anode and the electrolyte without anode functional layer. The LST/iLSBC/Pt half-cell achieved a power density of 377 mW/cm^2^ at 750 °C. These results proved that the activation, diffusion and ohmic resistances of the cell were effectively decreased using an ultrasonic spray pyrolysis process. The impregnated LSBC extended the TPB area and length, promoting the performance of the half-cell and also providing a beneficial and convenient unique reverse impregnation from electrolyte into porous electrode to fabricate a thin layer of electrolyte. Full cells with core-shell electrodes are prepared and studied in the next stage. The core-shell interface reaction mechanisms are also necessary investigation topics to control the electrochemical performance precisely. Advancing the core-shell technique up to applying double ions (H^+^ and O^2−^) conduction will realize low temperature solid oxide fuel (or electrolyzer) cells in the future.

## Figures and Tables

**Figure 1 polymers-13-02774-f001:**
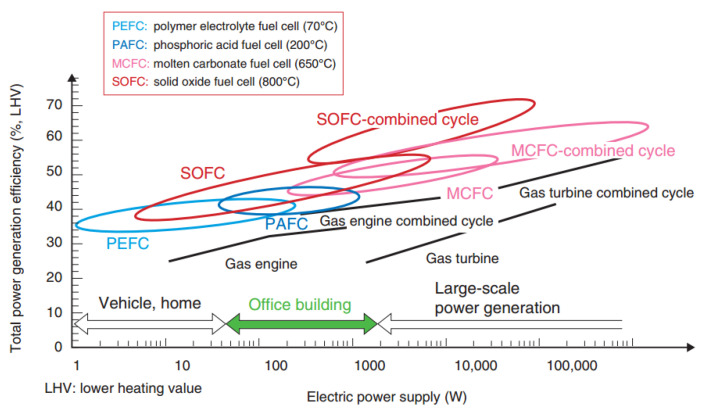
Total power generation efficiency of various fuel cells for the electric power output [4].

**Figure 2 polymers-13-02774-f002:**
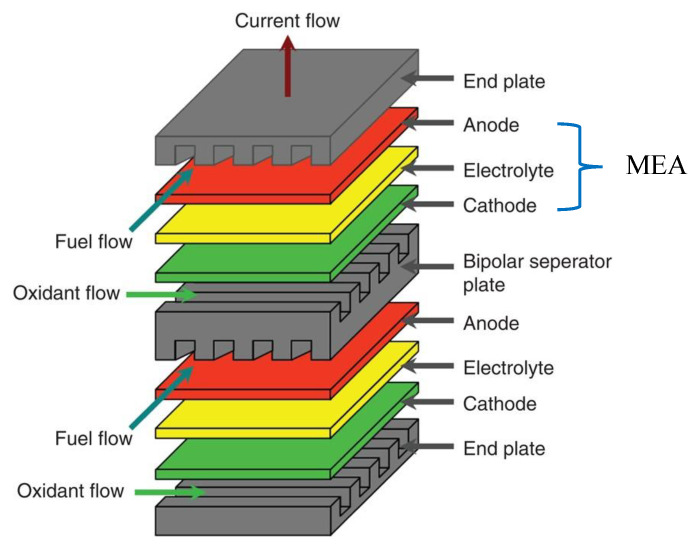
Planar solid oxide fuel cell stack with MEA indication [9].

**Figure 3 polymers-13-02774-f003:**
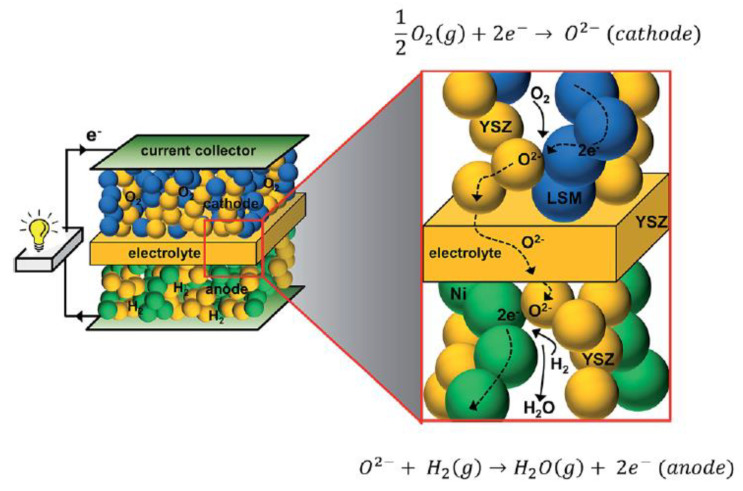
Schematic diagram for SOFC structure (left) and operation mechanism (right) [11].

**Figure 4 polymers-13-02774-f004:**
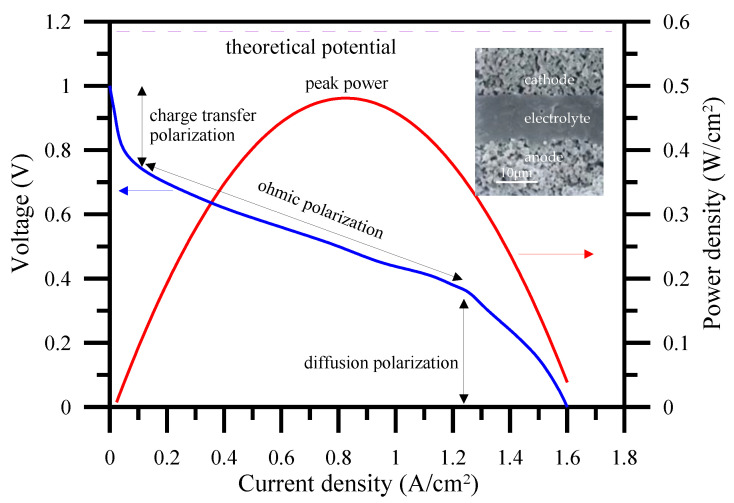
Typical electrochemical performance of a SOFC with I-V and I-P curves. The insect is a general microstructure of full cell (SOFC).

**Figure 5 polymers-13-02774-f005:**
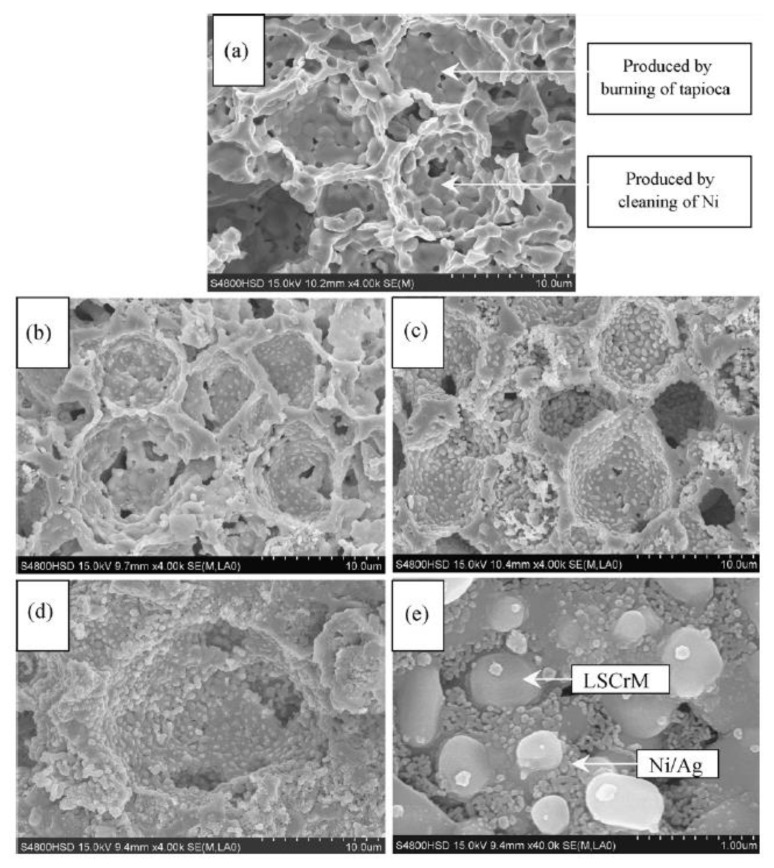
SEM micrographs of the cross-sections of: (**a**) pure YSZ anode backbone; (**b**) ~5 wt.% LSCrM-impregnated YSZ anode; (**c**) ~35 wt.% LSCrM-impregnated YSZ anode; (**d**,**e**) LSCrM/Ni/Ag (~32/6/2 wt.%) impregnated YSZ anode [33].

**Figure 6 polymers-13-02774-f006:**
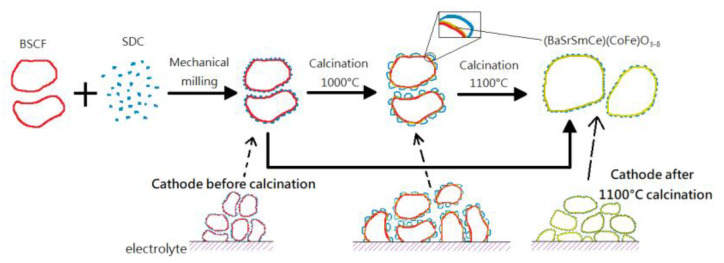
Schematic diagram for the sintering mechanism of BSCF + SDC composite cathode, redrawn from [34].

**Figure 7 polymers-13-02774-f007:**
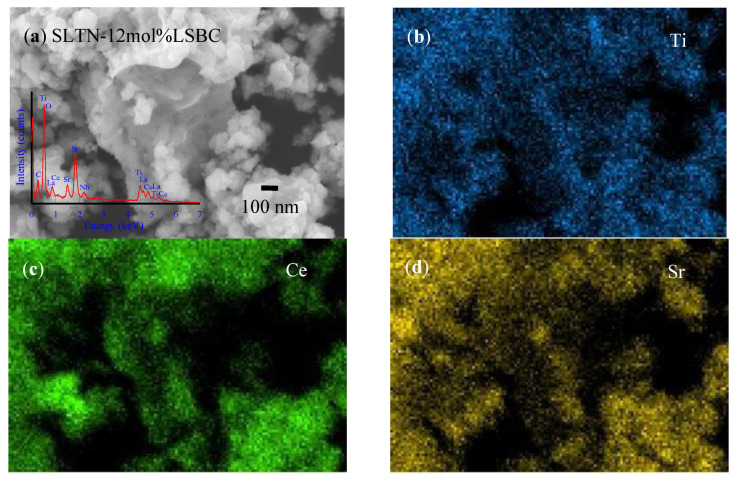
The FESEM images and EDS elemental analyses of the prepared core-shell anode powders, (**a**) FESEM image of SLTN-12 mol% LSBC, the inserted spectrum is its EDS analysis and the EDS elemental mappings on the SLTN-12 mol% LSBC, (**b**) Ti, (**c**) Ce and (**d**) Sr [56].

**Figure 8 polymers-13-02774-f008:**
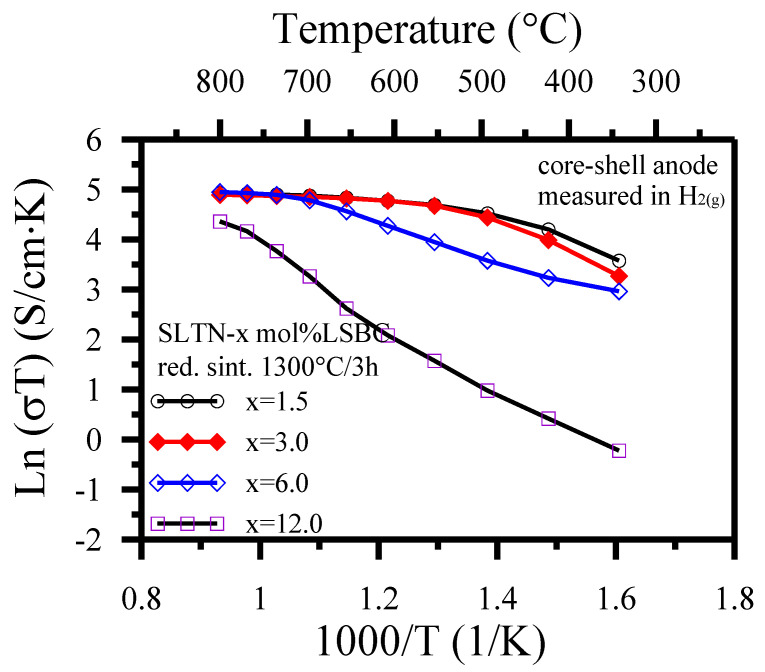
The conductivities of SLTN-x mole% LSBC core-shell anodes after 1300 °C/3 h sintering in activated carbon-reducing atmosphere as a function of measuring temperatures in H_2_ atmosphere.

**Figure 9 polymers-13-02774-f009:**
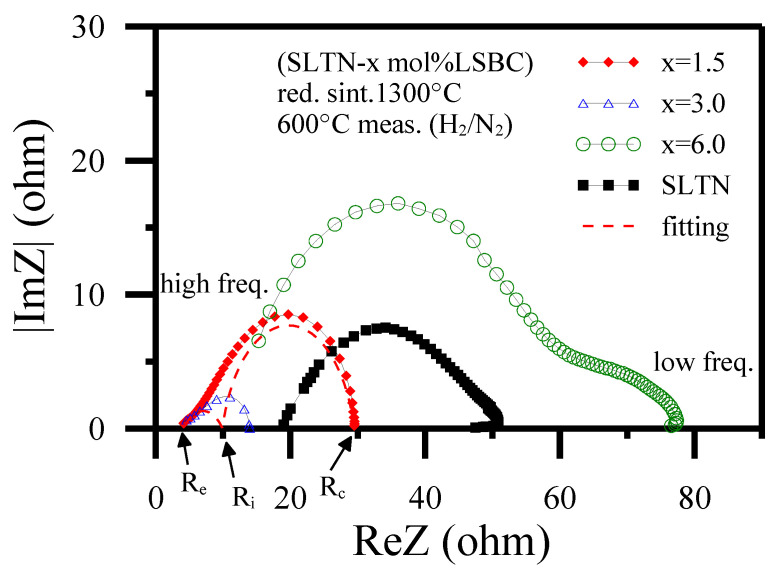
The AC impedance spectra for SLTN-x mol% LSBC core-shell anodes after 1300 °C/3 h sintering in activated carbon-reduction atmosphere then measured at 600 °C in H_2_/N_2_ atmosphere [56].

**Figure 10 polymers-13-02774-f010:**
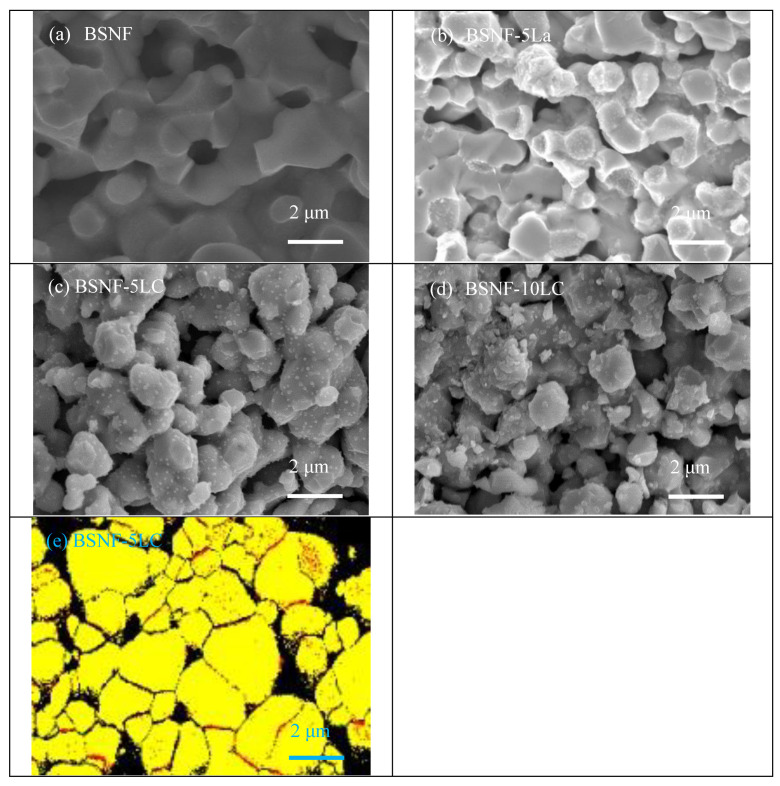
FESEM micrographs of (**a**) BSNF, (**b**) BSNF-5 La, (**c**) BSNF-5 LC and (**d**) BSNF-10 LC sintered at 1150 °C-6 h. EBSD image of (**e**) BSNF-5 LC. Red dots represent LC-species and yellow regions are BSNF grains [72].

**Figure 11 polymers-13-02774-f011:**
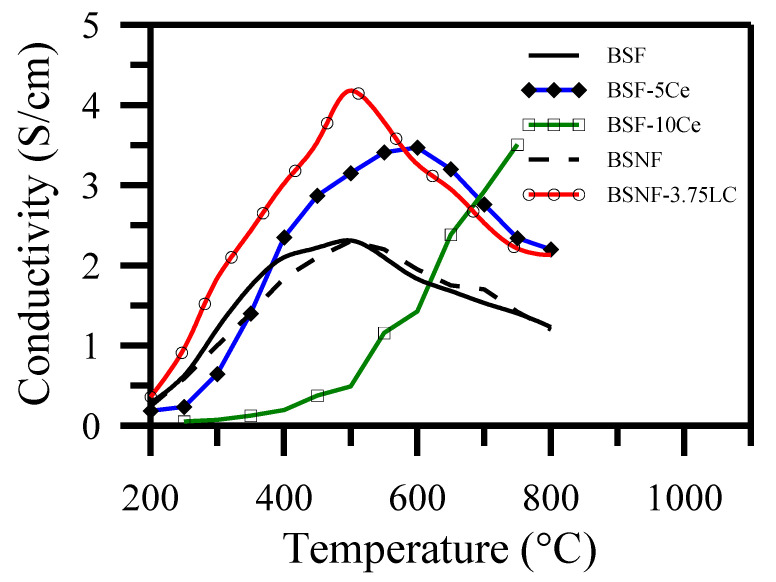
DC conductivity as function of temperature measured in air for BSF, BSF-y Ce (y = 5, 10), BSNF and BSF-3.75 LC [72].

**Figure 12 polymers-13-02774-f012:**
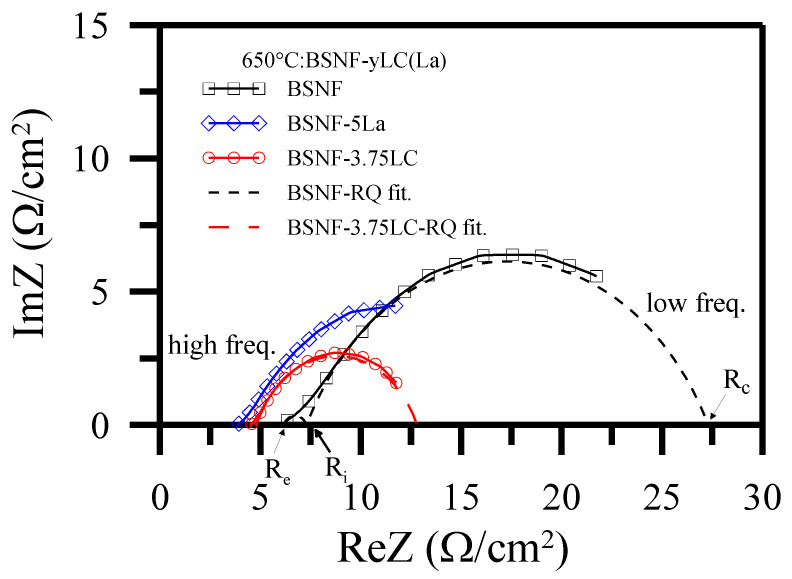
AC impedances of half-cells and RQ fittings for BSNF/LSBC/Pt and (BSNF-3.75 LC)/LSBC/Pt compared with (BSNF-5 La)/LSBC/Pt at 650 °C measurement [72].

**Figure 13 polymers-13-02774-f013:**
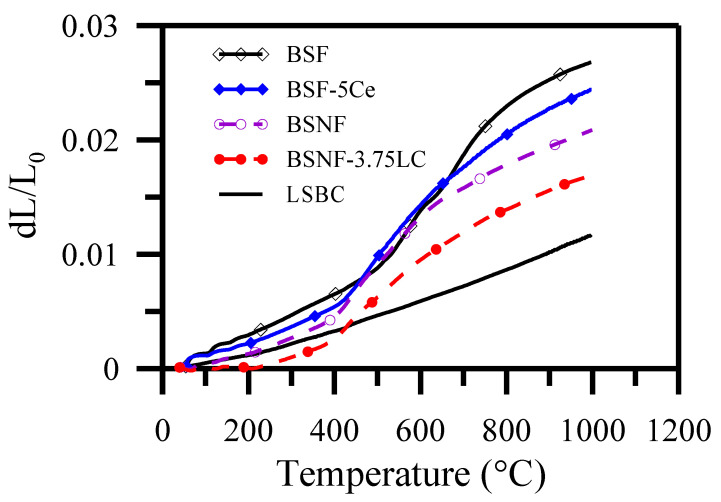
Comparisons of thermal expansion curves vs. temperatures for 1150 °C-6 h sintered BSF, BSF-5Ce, BSNF and BSNF-3.75LC; and 1500 °C-6 h sintered LSBC materials [72].

**Figure 14 polymers-13-02774-f014:**
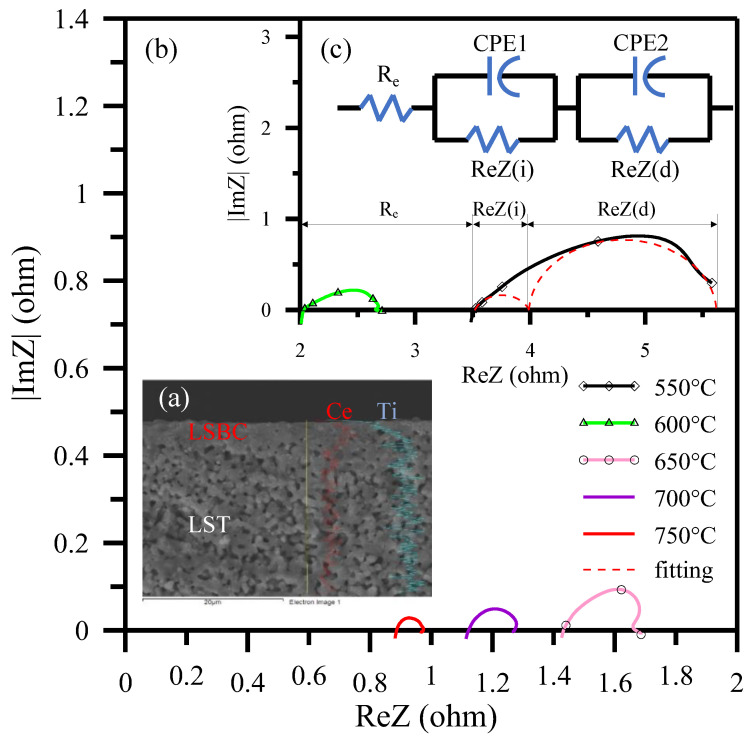
Co-fired iLSBC/LST half-cell prepared by ultrasonic spray pyrolysis and measured at a temperature of 550–750 °C for (**a**) FESEM cross-section of iLSBC/LST with line scanning marking Ce (red color) and Ti (light blue color) elements, AC impedances among (**b**) 650–750 °C and (**c**) 550–600 °C and RQ fitting for 550 °C.

**Table 1 polymers-13-02774-t001:** Functional comparisons of anode-supported SOFCs with thin layer of ceria-based electrolyte operated in H_2_ (or humidified H_2_) as fuel and air as oxidant atmosphere (cited from the literatures and this work).

Cell Configuration	Electrolyte Prepared Method	Electrolyte Thickness (μm)	Electrolyte Prepared Temperature (°C)	Temperature of Co-Fired with Anode (°C)	Peak Power Density (mW/cm^2^)				Ref.
					600 °C	650 °C	700 °C	750 °C	
S.S/Ni + SDC/SCSZ/LSM + SDC/S.S. *	infiltration	7	Tape casting	1350			900		[98]
Ni + GDC/GDC/SSC + GDC	dry pressing	20	600	1350	400				[99]
Ni + SDC/SDC/SSC + SDC	screen printing	30		1350	397				[100]
Ni + GDC/GDC/LSCF + GDC	dip coating	10	commercial	1450	300				[101]
Ni + GDC/GDC/LSCF	spin coating + co-pressing	4	commercial	1300	771				[102]
Ni + GDC/GDC/LSCF	spin coating	19	700	1350	386	492			[103]
Ni + GDC/GDC/LSCF + GDC	spray coating	10	commercial	1450	578				[90]
Ni + Ru + GDC/GDC/SSC	spin coating	40	commercial	1500	250				[104]
Ni + YSZ/GDC/SSC + SDC	repeated spray pyrolysis	2.08	precursor	600	280	350			[105]
LST/LSBC/Pt (half-cell)	ultrasonic spray pyrolysis	2.75	450	1350	142	240	325	377	our work

* S.S.: porous stainless steel; SCSZ: 10ScSZ + 1 mol% CeO_2_ electrolyte.

## Data Availability

All data are offered by the authors by reasonable request and the novel core-shell electrodes are available from the authors.

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
