# Peer review of "An Overview on the Novel Core-Shell Electrodes for Solid Oxide Fuel Cell (SOFC) Using Polymeric Methodology"

_polymers, 2021, doi:10.3390/polym13162774_

Round 1

Reviewer 1 Report

Dear Author,

This paper is about the Solid Oxide Fuel Cell (SOFC) using Polymeric Methodology. I presented my review in the attachment file.

The article is well written and seems to be free of technical errors.

These all the mistakes and errors can be corrected that I can accept this paper. This paper is a minor revision.

Reviewer

Author Response

Response to Reviewer 1 Comments

The authors are grateful to the reviewers whose comments have helped us improve the manuscript. Thanks Reviewers for their time and effort for the present manuscript. I appreciate his/her comments that o work presents “The article is well written and seems to be free of technical errors.” And “These all the mistakes and errors can be corrected that I can accept this paper. This paper is a minor revision.”. We have reconstructed and reorganized the whole manuscript carefully. And in order to improve the quality of the paper, the whole manuscript has been checked carefully to avoid any grammar or syntax error. Meanwhile, some statements have been rephrased or refined to improve the language usage in the manuscript. We have accounted and discussed the phenomenon more details in the revised version. We have added more comments to explain the physical effects of the plots. The review comments and descriptions of revisions are listed as following. We have included a separate copy of the revised paper in which we highlighted all the modifications made according to all the reviewers’ comments in RED colour. We believe that the present paper is now acceptable for publication.

Point 1: The title is clear and it is adequate to the content of the article.

Response 1: Thanks for the reviewer’s comment.

Point 2: The novelty and originality of this work is clear. However, some correction should be in the Abstract that can be decreased and put these sentence into Introduction section. You should give the result of the study and aim of the study, clearly. For instance, “The result of the study is to be 50 kW……..” and “The aim of the study is ………..”

Response 2: We appreciate the abstract comments for us to more focuses on the paper contribution clearly. The detailed written and described contents are in the revised version.

Point 3: The purpose or purported significance of the article is explicitly stated.

Response 3: Thanks for the reviewer’s comment.

Point 4: Result and Discussion is clear. Please design your section as you can prepare this section comparison of the previous studied according to literature. Because references quantity is inadequate.

Response 4: The reviewer gives the important suggestions for our result and discussion to improve the fulfil overview in our work and comparisons with other researchers’ studies.

We have rewritten and rearranged the sections in the revised version. The detailed paper reviews and comparisons with our results are also described and give suggestions in the text. The total references are 108 papers in the revised version. We hope the references weight fitting the requirement of our kind reviewer.

Point 5: The literature review and research study methods are explained clearly. Please see some advices as follow comments. Because this journal is a Science Index Expanded Journal.

Your study includes polymer method with SEM analysis, so you should add this reference to your required sections (also you can increase previous studies) as follows:

- The preparation, characterization and antibacterial properties of chitosan/pectin silver nanoparticle films, Polymer Bulletin, 1-18 (2021). doi: 10.1007/s00289-021-03667-0.

Author, who should give reference for the fuel cell, may provide other fuel cell as follows:

- The novel and innovative design with using H2 fuel of PEM fuel cell: Efficiency of thermodynamic analyze. Fuel 302 (121109), 1-11. Doi: 10.1016/j.fuel.2021.121109.

- Energy and exergy analyze of PEM fuel cell: A case study of modeling and simulations. Energy, 143, 284-294.(2018). Doi: 10.1016/j.energy.2017.10.102.

Response 5: Thank you for reviewer comments and provide the relative references. We have used those references in the introduction and experimental sections. Other useful references are added and discussed in the revised version. The followings are the referenced papers in the revised text indicated the reference number.

  1. Taner T. Energy and exergy analyze of PEM fuel cell: A case study of modeling and simulations, Energy, 2018, 143, 284-294.
  2. Taner T. The novel and innovative design with using H2 fuel of PEM fuel cell: Efficiency of thermodynamic analyze. Fuel, 2021, 302, 121109 (11 pages).
  3. Gulen Oytun Akalin, Oznur Oztuna Taner, Tolga Taner. The preparation, characterization and antibacterial properties of chitosan/pectin silver nanoparticle films. Polym. Bull. (2021). https://doi.org/10.1007/s00289-021-03667-0
  4. Dhanapal A, Sasikala P, Rajamani L, Kavitha V, Yazhini G, Banu MS. Edible films from polysaccharides. Food Sci Qual Manag, 2012, 3, 2224–6088
  5. Cazon P, Velazquez G, Ramirez JA, Vazquez M (2017) Polysaccharide-based films and coatings for food packaging: a review. Food Hydrocoll, 2017, 68, 136–148.
  6. Altun T. Preparation and application of glutaraldehyde cross-linked chitosan coated bentonite clay capsules: chromium (VI) removal from aqueous solution. J. Chil. Chem. Soc., 2020, 65(2), 4790–4797.
  7. dos Santos DS, Goulet PJ, Pieczonka NP, Oliveira ON, Aroca RF. Gold nanoparticle embedded, self-sustained chitosan films as substrates for surface-enhanced Raman scattering. Langmuir, 2004, 20(23), 10273–10277.
  8. Cagri A, Ustunol Z, Ryser ET. Antimicrobial edible films and coatings. J. Food Prot., 2004, 67, 833–848.

Point 6: Supplements (tables, charts, pictures and drawings) are necessary and clear.

Response 6: Thank you for your suggestions. We have redrawn or add figures in the revised version such as Figures 4, 6, 9, and 12-14, also table1 for comparisons with other papers. Those are expected to improve the revised paper with more clearly and readably.

Point 7: The conclusions is accurate and supported by the content.

Response 7: Thanks for the reviewer’s comment.

Point 8: The references are comprehensive and appropriate.

Response 8: Thanks for the reviewer’s comment.

Point 9: In order to give our readers a sense of continuity, I can encourage you to identify journal publications of similar research in your papers. You should make a literature check of the papers published in recent years (2019, 2020 and even 2021) and relate the content of relevant papers to the results and findings presented in your publication.

Response 9: It’s a good suggestion. Thank you very much. We have increased references to 108 papers. The relative papers in 2019-2021 are listed in the following. We also use these papers for discussion and comparisons in words, figures and table in our revised version. Please check the contents in the revised paper.

  1. Taner T. The novel and innovative design with using H2 fuel of PEM fuel cell: Efficiency of thermodynamic analyze. Fuel, 2021, 302, 121109 (11 pages).
  2. Ng K.H., Rahman H.A., Somalu M.R. Review: Enhancement of composite anode materials for low-temperature solid oxide fuels. Int. J. Hydrogen energy, 2019, 44 [58], 30692-30704.
  3. Abd Aziz A.J., Baharuddin N.A., Somalu M.R., Muchtar A. Review of composite cathodes for intermediate-temperature solid oxide fuel cell applications, Ceram. Int., 2020, 46, 23314-23325.
  4. He A.A., Jiang S.P. Electrode/electrolyte interface and interface reactions of solid oxide cells: Recent development and advances, Prog. Natural Sc.: Mater. Int., 2021, 31, 341–372.
  5. Shekhar R.S., Bertei A., and Monder D.S. Structure - properties - performance: modelling a solid oxide fuel cell with infiltrated electrodes. J. Electrochem. Soc., 2020, 167, 084523.
  6. Vijay P., Tadé M.O., Shao Z.P. Model based evaluation of the electrochemical reaction sites in solid oxide fuel cell electrodes. Int. J. Hydrogen Energy, 2019, 44, 8439-8459.
  7. Yashima, M.; and Takizawa, T. Atomic displacement parameters of ceria doped with rare-earth oxide Ce0.8R0.2O1.9 (R = La, Nd, Sm, Gd, Y, and Yb) and correlation with oxide-ion conductivity. J. Phys. Chem. C, 2020, 114, 2385–2392.
  8. Shuai He, San Ping Jiang. Electrode/electrolyte interface and interface reactions of solid oxide cells: Recent development and advances, Progress in Natural Science: Materials International, 2021, 31, 341–372
  9. L. Zhang, X. Li, L. Zhang, H. Cai, J. Xu, L. Wang, W. Long, Improved thermal expansion and electrochemical performance of La0.4Sr0.6Co0.9Sb0.1O3-δ-Ce0.8Sm0.2O1.9 composite cathode for IT-SOFCs, Solid State Sci.,2019, 91, 126–132.
  10. Gulen Oytun Akalin, Oznur Oztuna Taner, Tolga Taner. The preparation, characterization and antibacterial properties of chitosan/pectin silver nanoparticle films. Polym. Bull. (2021).
  11. Altun T. Preparation and application of glutaraldehyde cross-linked chitosan coated bentonite clay capsules: chromium (VI) removal from aqueous solution. J. Chil. Chem. Soc., 2020, 65(2), 4790–4797.
  12. S.P. Jiang, Development of lanthanum strontium cobalt ferrite perovskite electrodes of solid oxide fuel cells – a review, Int. J. Hydrogen Energy, 2019, 44, 7448–7493.
  13. Fu-Yin Ko, Te-Wei Chiu, Rudder Wu, Tai-Cheng Chen, Horng-Yi Chang. Thin Layer Electrolyte Impregnation into Porous Anode Supported Fuel Cell by Ultrasonic Spray Pyrolysis. Int. J. Hydrogen Energy, 2021, 46[31], 16708-16716.
  14. Seo H, Kishimoto M, Ding C, Iwai H, Saito M, Yoshida H. Improvement in the electrochemical performance of anode supported solid oxide fuel cells by meso- and nanoscale structural modifications. Fuel Cell, 2020, 20, 570-579.
  15. Lu YC, Gasper P, Nikiforov AY, Pal UB, Gopalan S, Basu SN. Co-infiltration of nickel and mixed conducting Gd0.1Ce0.9O2-d and La0.6Sr0.3Ni0.15Cr0.85O3-d phases in Ni-YSZ anodes for improved stability and performance. JOM 2019, 71, 3835-3847.
  16. Tian YT, Guo X, Wu PP, Zhang X, Nie ZQ. Preparation and evaluation of Ni-based anodes with straight open pores for solid oxide fuel cells. J. Alloys Compd., 2020, 817, 153244.
  17. Emir Dogdibegovic, Ruofan Wang, Grace Y. Lau, M.C. Tucker. High performance metal-supported solid oxide fuel cells with infiltrated electrodes. J. Power Sources, 2019, 410–411, 91–98.
  18. Takahashi S, Sumi H, Fujishiro Y. Development of cosintering process for anode supported solid oxide fuel cells with gadolinia doped ceria/lanthanum silicate bi-layer electrolyte. Int. J. Hydrogen Energy, 2019, 44, 23377-23383.

Point 10: The manuscript contains new and original principles, concepts, methods and applications.

Response 10: Thanks for the reviewer’s comment.

Reviewer 2 Report

Greetings, Editor thank you for providing me with the opportunity to review the article. I reviewed the article by Wang et al. (Polymers-1288645). Apologies to say that the present article arrangement and content are very rough and irregular. It does not meet the Polymers standard. Therefore, I reject in present form.

  1. The present information is very general and commonly available in previous literature so what is need for this study?
  2. 34% work is directly copied work. The plagiarism should be below than 20%. The authors efforts seems very little in present work.
  3. The abstract should be clearly on main idea. Please write few lines on introduction of work then your objective and conclusion.
  4. There are several statements need a references. Each reference needs to be properly addressed. Please revise your paper accordingly since same issue occurs on several spots in the paper.
  5. The novelty of the work must be clearly addressed and discussed, compare existing research findings and highlight novelty.
  6. Research gap should be delivered on more clear way with directed necessity for the present study.
  7. The main objective of the work must be written on the more clear and more concise way at the end of introduction section. It seems weird in present form.
  8. Section 5 should be renamed by Conclusion and Future perspectives. Conclusion section is missing some perspective related to the future research work, quantify main research findings, highlight relevance of the work with respect to the field aspect.
  9. To avoid grammar and linguistic mistakes, major level English language should be thoroughly checked.

Overall, it seems that authors did not put sincere efforts, so it is difficult to give corrections in present form. I encourage authors to revise the article properly and re-submit again.

Author Response

Response to Reviewer 2 Comments

The authors are grateful to the reviewers whose comments have helped us improve the manuscript. Thanks Reviewers for their time and efforts for the present manuscript. We have reconstructed and reorganized the whole manuscript carefully. And in order to improve the quality of the paper, the whole manuscript has been checked carefully to avoid any grammar or syntax error. Meanwhile, some statements have been rephrased or refined to improve the language usage in the manuscript. We have accounted and discussed the phenomenon more details in the revised version. We have added more comments to explain the physical effects of the plots. The review comments and descriptions of revisions are listed as following. We have included a separate copy of the revised paper in which we highlighted all the modifications made according to all the reviewers’ comments in RED colour. The authors believe that the present paper is now acceptable for publication.

Point 1: The present information is very general and commonly available in previous literature so what is need for this study?

Response 1: Thanks for the reviewer’s comment. The manuscript is changed to “review paper”. The illustrations about the importance and differences of our work with previous literatures are as the follows:

Composite electrodes by mixing ionic and electronic conducting materials are used to improve electrodes performance. Such composite electrodes including anode and cathode help to enhance the properties of mixed electronic–ionic conductors and the inter-component compatibility [7, 25-31].

Some composite electrode examples are also presented as following. The Ni-BZCY/SDC/BSCF cell that interfacial reaction can be manipulated to form a secondary phase at anode Ni-BZCY and SDC that with electronic conductor to benefit of cell performance and power output [35]. The secondary phases nickel aluminate spinel (NiAl2O4) and zirconium titanite (Zr5Ti7O24) formation by infiltrating a small amount of aluminium titanite (ALT, ~4 wt%) into the Ni-YSZ anode scaffold were found to suppress Ni coarsening and expand the electrode’s TPBs [36, 37].

Another TPBs extension method is an introduction of the anode functional layer (AFL) with fine microstructure at the anode/electrolyte interface to increase the TPB length and to restrain the activation polarization for hydrogen oxidation [38-40]. This kind of technique meets a trade-off between the enhancement of electrochemical performance due to the increasing TPB and the decrease of performance due to the increased gas diffusion resistance.

The core-shell is not a new terminology. It has been used for ceramics in several studies [51-53]. However, in our work, the core-shell electrodes pre-formed by chelating solution to extend TPBs was appreciated as the beneficial developing technology in future for SOFC [13]. The main objective in this work is then to demonstrate the progress of core-shell electrodes by our efforts to provide a feasible, convenient, cost-effective and time-saving TPBs extension technique. The proposed core-shell electrodes from fabrication by chemical chelating and solution coating processes to electrical and electrocatalytic characterization are informed in this paper. Furthermore, the recent developing pseudo-core-shell anode by thin electrolyte impregnation is introduced and compares with other ITSOFC researches. The pseudo-core-shell or inverse impregnation is further expected to apply to double ions (H+ and O2-) conducting low temperature fuel cells.

The above illustrations are also written in the revised version. The numbers are reference number in the revised paper.

Point 2: 34% work is directly copied work. The plagiarism should be below than 20%. The authors efforts seems very little in present work.

Response 2: Thanks for the reviewer’s comment. The manuscript is changed to “review paper”. We have payed heavy efforts on our revised version. The total article numbers are up to 108 reference papers. We have rewritten and rearranged the sections in the revised version. The detailed paper reviews and comparisons with our results are also described and give suggestions in the text. The authors hope the cited our published contents in the revised version to reach to less than 20%. Please check the revised contents and find our sincerely efforts.

Point 3: The abstract should be clearly on main idea. Please write few lines on introduction of work then your objective and conclusion.

Response 3: Thank you for your kind suggestions. The abstract has been rewritten to emphasized and clarify the main idea of our work clearly. The main objective in this work is to demonstrate the progress of core-shell electrodes by our efforts to provide a feasible, convenient, cost-effective and time-saving TPBs extension technique. The proposed (pseudo-) core-shell electrode particles were pre-formed by chemical chelating and solution coating processes to extend TPBs and then enhanced the electrodes electrocatalytic activity to promote the SOFC performance. The pseudo-core-shell or inverse impregnation is further expected to apply to double ions (H+ and O2-) conducting for low temperature fuel cells.

Point 4: There are several statements need a references. Each reference needs to be properly addressed. Please revise your paper accordingly since same issue occurs on several spots in the paper.

Response 4: This is an important comment. The authors have rewritten and rearranged the sections in the revised version. The detailed paper reviews and comparisons with our results are also described and give suggestions in the text. The total numbers are 108 reference papers in the revised version. We hope the references weight and full discussion and comparisons fitting the requirement of our kind reviewer.

Point 5: The novelty of the work must be clearly addressed and discussed, compare existing research findings and highlight novelty.

Response 5: Thank you for your excellent hints and suggestions. The pre-formed core-shell electrode particles prepared by chelating solution and co-fired with electrolyte to extend TPBs was appreciated as the beneficial developing technology in future for SOFC [13]. This is a feasible, convenient, cost-effective and time-saving TPBs extension technique. Recently developed pseudo-core-cell technique further provides electrode impregnation and dense electrolyte growth successfully in one process. It can be expected to develop double ions (H+ and O2-) conducting low temperature fuel cells.

Point 6: Research gap should be delivered on more clear way with directed necessity for the present study.

Response 6: The reviewer gives a key comment to highlight our studies. No matter what mixing, infiltration and functional layer methods to extend the TPBs in SOFC electrodes, they require additional processing and cost. Repeated impregnation by capillary action may decrease the efficiency of the reforming and electrocatalysis, possibly by hindering and even blocking the fuel gas path through the porous electrode structure due to the serious agglomeration of impregnated particles. Our (pseudo-) core-cell structure preparation provides a feasible, convenient, cost-effective and time-saving TPBs extension technique. Recently developed pseudo-core-cell technique further provides electrode impregnation and dense electrolyte growth successfully in one process. It can be expected to develop double ions (H+ and O2-) conducting low temperature fuel cells.

Point 7: The main objective of the work must be written on the more clear and more concise way at the end of introduction section. It seems weird in present form.

Response 7: Thank you for this comment. We have rewritten and rearranged the sections in the revised version. The detailed paper reviews and comparisons with our results are also described and give suggestions in the text. The main objective is also presented at the end of introduction as following:

The main objective in this work is then to demonstrate the progress of core-shell electrodes by our efforts to provide a feasible, convenient, cost-effective and time-saving TPBs extension technique. The proposed core-shell electrodes from fabrication by chemical chelating and solution coating processes to electrical and electrocatalytic characterization are informed in this paper. Furthermore, the recent developing pseudo-core-shell anode by thin electrolyte impregnation is introduced and compares with other ITSOFC researches. The pseudo-core-shell or inverse impregnation is further expected to apply to double ions (H+ and O2-) conducting low temperature fuel cells. The authors hope the new written research objective can fit the reviewer requirements.

Point 8: Section 5 should be renamed by Conclusion and Future perspectives. Conclusion section is missing some perspective related to the future research work, quantify main research findings, highlight relevance of the work with respect to the field aspect.

Response 8: Thank you for your kind suggestions. The conclusion has been renamed by Conclusion and Future respectives. Please check the revised version.

A recent pseudo-core-shell technique by our group is also integrated in the revised version. The ultrasonic spray pyrolyzed LSBC electrolyte droplets impregnated into LST porous anode to form a pseudo-core-shell and continued the deposition as to resulting in a fired thickness of 2.75 mm dense LSBC electrolyte. This pseudo-core-shell anode substrate improved contact between the anode and the electrolyte without anode functional layer. The LST/iLSBC/Pt only half-cell achieved a power density of 377 mW/cm2 at 750 °C. These results proved that the activation, diffusion, and ohmic resistances of the cell were effectively decreased using an ultrasonic spray pyrolysis process. The impregnated LSBC extended the TPB area and length, promoting the performance of the half-cell and also providing a beneficial and convenient unique reverse impregnation from electrolyte into porous electrode to fabricate a thin layer of electrolyte. Full cells with core-shell electrodes are prepared and studied in next stage. The core-shell interface reaction mechanisms are also necessary investigation topics to control the electrochemical performance precisely. Advancing the core-shell technique up to applying double ions (H+ and O2-) conduction will realize low temperature solid oxide fuel (or electrolyzer) cells in the future.

Point 9: To avoid grammar and linguistic mistakes, major level English language should be thoroughly checked.

Response 9: Thank to reviewer suggestion about language modification. In order to improve the quality of the paper, the whole manuscript has been checked carefully to avoid any grammar or syntax error. Meanwhile, some statements have been rephrased or refined to improve the language usage in the manuscript.

Reviewer 3 Report

From the content of the manuscript, it is understood that it is a Review of literature because the figures are taken from articles already published. The framing of the manuscript was done in Article (original), and the authors talk about newly synthesized material. The authors are asked to explain the status of the paper and to remedy the confusion.

The whole manuscript is a little confusing for the reader, because it is not clear whether the authors discuss comparative data from the literature, highlighting materials with superior qualities, or have synthesized a new material for which comparison is made with literature data. The whole manuscript needs to be restructured to be clearer. If the study is original, why are the figures taken from other publications? The authors must mention from the beginning the purpose of the study, what they set out to do and what is the novelty and originality of the study. These aspects do not appear in the current version of the manuscript and should be explained by the authors.

Are the data in the 2 tables original or taken from the literature? If the last option is valid, the corresponding citation must be considered.

Some figures have poor quality, so they should be replaced with improved quality figures. An improved quality figure would be needed instead of Figure 1. In Figure 2 , Figure 5, Figure 6 and Figure 11, the text is blurry and difficult to read. There are missing parts of Figure 9. Authors are kindly asked to revise this aspect

The manuscript must be read carefully because there are errors of expression, typing, and language that need to be revised.

In this form, the manuscript cannot be recommended for publication in Polymers. My proposal is a major revision.

Author Response

Response to Reviewer 3 Comments

The authors are grateful to the reviewers whose comments have helped us improve the manuscript. Thanks Reviewers for their time and efforts for the present manuscript. We appreciate his/her comments that my work presents “My proposal is a major revision.”. We have reconstructed and reorganized the whole manuscript carefully. And in order to improve the quality of the paper, the whole manuscript has been checked carefully to avoid any grammar or syntax error. Meanwhile, some statements have been rephrased or refined to improve the language usage in the manuscript. The authors have accounted and discussed the phenomenon more details in the revised version. We have added more comments to explain the physical effects of the plots. The review comments and descriptions of revisions are listed as following. I have included a separate copy of the revised paper in which we highlighted all the modifications made according to all the reviewers’ comments in RED colour. We believe that the present paper is now acceptable for publication.

Point 1: From the content of the manuscript, it is understood that it is a Review of literature because the figures are taken from articles already published. The framing of the manuscript was done in Article (original), and the authors talk about newly synthesized material. The authors are asked to explain the status of the paper and to remedy the confusion.

Response 1: Thanks for the reviewer’s comment. The manuscript belongs to the review paper not the original type. The authors have rewritten and rearranged the sections in the revised version. The importance of our work and differences with previous literatures are described as the following to explain the progress and highlights of our work.

Composite electrodes by mixing ionic and electronic conducting materials are used to improve electrodes performance. Such composite electrodes including anode and cathode help to enhance the properties of mixed electronic–ionic conductors and the inter-component compatibility [7, 25-31].

Some composite electrode examples are also presented as following. The Ni-BZCY/SDC/BSCF cell that interfacial reaction can be manipulated to form a secondary phase at anode Ni-BZCY and SDC that with electronic conductor to benefit of cell performance and power output [35]. The secondary phases nickel aluminate spinel (NiAl2O4) and zirconium titanite (Zr5Ti7O24) formation by infiltrating a small amount of aluminium titanite (ALT, ~4 wt%) into the Ni-YSZ anode scaffold were found to suppress Ni coarsening and expand the electrode’s TPBs [36, 37].

Another TPBs extension method is an introduction of the anode functional layer (AFL) with fine microstructure at the anode/electrolyte interface to increase the TPB length and to restrain the activation polarization for hydrogen oxidation [38-40]. This kind of technique meets a trade-off between the enhancement of electrochemical performance due to the increasing TPB and the decrease of performance due to the increased gas diffusion resistance.

The core-shell is not a new terminology. It has been used for ceramics in several studies [51-53]. However, in our work, the core-shell electrodes pre-formed by chelating solution to extend TPBs was appreciated as the beneficial developing technology in future for SOFC [13]. The main objective in this work is then to demonstrate the progress of core-shell electrodes by our efforts to provide a feasible, convenient, cost-effective and time-saving TPBs extension technique. The proposed core-shell electrodes from fabrication by chemical chelating and solution coating processes to electrical and electrocatalytic characterization are informed in this paper. Furthermore, the recent developing pseudo-core-shell anode by thin electrolyte impregnation is introduced and compares with other ITSOFC researches. The pseudo-core-shell or inverse impregnation is further expected to apply to double ions (H+ and O2-) conducting low temperature fuel cells.  The numbers are reference numbers in the revised paper.

Point 2: The whole manuscript is a little confusing for the reader, because it is not clear whether the authors discuss comparative data from the literature, highlighting materials with superior qualities, or have synthesized a new material for which comparison is made with literature data. The whole manuscript needs to be restructured to be clearer. If the study is original, why are the figures taken from other publications? The authors must mention from the beginning the purpose of the study, what they set out to do and what is the novelty and originality of the study. These aspects do not appear in the current version of the manuscript and should be explained by the authors.

Response 2: Thank you for reviewer to emphasize the novelty and main idea for our work. We have rewritten and rearranged the sections in the revised version. The detailed paper reviews and comparisons with our results are also described and give suggestions in the text. The total references are 108 papers in the revised version.

This paper is to demonstrate the progress of core-shell electrodes by our efforts to provide a feasible, convenient, cost-effective and time-saving TPBs extension technique. The proposed core-shell electrodes from fabrication by chemical chelating and solution coating processes to electrical and electrocatalytic characterization are informed in this paper. Furthermore, the recent developing pseudo-core-shell anode by thin electrolyte impregnation is introduced and compares with other ITSOFC researches.

The ultrasonic spray pyrolyzed LSBC electrolyte droplets impregnated into LST porous anode to form a pseudo-core-shell and continued the deposition as to resulting in a fired thickness of 2.75 mm dense LSBC electrolyte. This psudo-core-shell anode substrate improved contact between the anode and the electrolyte without anode functional layer. The LST/iLSBC/Pt half-cell achieved a power density of 377 mW/cm2 at 750 °C. These results proved that the activation, diffusion, and ohmic resistances of the cell were effectively decreased using an ultrasonic spray pyrolysis process.

The impregnated LSBC extended the TPB area and length, promoting the performance of the half-cell and also providing a beneficial and convenient unique reverse impregnation from electrolyte into porous electrode to fabricate a thin layer of electrolyte. The pseudo-core-shell or inverse impregnation is further expected to apply to double ions (H+ and O2-) conducting low temperature fuel cells.

Point 3: Are the data in the 2 tables original or taken from the literature? If the last option is valid, the corresponding citation must be considered.

Response 3: Thank you for the reviewer’s findings. The authors have replaced the 2 Tables by Figures 11 and 13 in the revised version and give detailed illustrations and discussion.

Point 4: Some figures have poor quality, so they should be replaced with improved quality figures. An improved quality figure would be needed instead of Figure 1. In Figure 2 , Figure 5, Figure 6 and Figure 11, the text is blurry and difficult to read. There are missing parts of Figure 9. Authors are kindly asked to revise this aspect.

Response 4: Thank you for your checking in details. The Figure 1 is a good presentation for comparisons of various fuels cells. Basically, the resolution is acceptable and distinguishable. Please allow to keep the original citation. The Figure 2 has been replaced by a new cited Figure 2 in the revised version. The Figure 5 has been redrawn clearly as Figure 6 in the revised version to indicate the mechanism with precisely. The Figure 6 is reassembled to Figure 7 in the revised version for more readable. The Figure 11 is redrawn to Figure 12 in the revised version with more clear markings and corresponding illustration is written precisely in the text. The missing markings (a)~(e) are corrected and becomes Figure 10 in the revised version.

Point 5: The manuscript must be read carefully because there are errors of expression, typing, and language that need to be revised.

Response 5: Thank you for suggesting the typing and language expression errors correction. The whole manuscript has been checked carefully to avoid any grammar or syntax error. Meanwhile, some statements have been rephrased or refined to improve the language usage in the manuscript.

Round 2

Reviewer 2 Report

Dear Authors i really appreciate your efforts. The present form is acceptable for publication.

Reviewer 3 Report

The revised form of the manuscript has a much better quality compared to the initial version. The authors considered the suggestions and completed and modified the manuscript.
The quality of the figures has been improved, the bibliographic references completed and the data presented in a critical manner, suitable for review article.
In this form, I can recommend this study for publication in the Polymers journal.